# Full-Fat Black Soldier Fly Larvae Meal in Diet for Tambaqui, *Colossoma macropomum*: Digestibility, Growth Performance and Economic Analysis of Feeds

**DOI:** 10.3390/ani13030360

**Published:** 2023-01-20

**Authors:** Driely Kathriny Monteiro dos Santos, Odair Rodrigues de Freitas, Cesar Augusto Oishi, Flávio Augusto Leão da Fonseca, Giuliana Parisi, Ligia Uribe Gonçalves

**Affiliations:** 1Programa de Pós-Graduação em Aquicultura, Universidade Nilton Lins, Av. Prof. Nilton Lins 3259, Parque das Laranjeiras, Manaus 69065-030, Amazonas, Brazil; 2Faculty of Agricultural Sciences, Universidade Federal do Amazonas, Av. General Rodrigo Octavio Jordão Ramos, 1200-Coroado I, Manaus 69067-005, Amazonas, Brazil; 3Instituto Nacional de Pesquisas da Amazônia, Av. André Araújo 2936, Manaus 69060-001, Amazonas, Brazil; 4Instituto Federal de Educação, Ciência e Tecnologia do Amazonas-Campus Zona Leste, Av. Cosme Ferreira, São José Operário, Manaus 69083-000, Amazonas, Brazil; 5Dipartimento di Scienze e Tecnologie Agrarie, Alimentari, Ambientali e Forestali, Università degli Studi di Firenze, Via delle Cascine 5, 50144 Firenze, Italy

**Keywords:** aquafeed, *Hermetia illucens*, insect meal

## Abstract

**Simple Summary:**

The black soldier fly (BSF, *Hermetia illucens*) is one of the most studied insects for use in feed, whose larvae (BSFL) can be used to feed fish. In this study, the inclusion of 10.5% wholemeal BSFL meal in the diet of tambaqui (a neotropical fish) resulted in higher growth than fish fed conventional ingredients. The use of wholemeal BSFL meal is more advantageous because it has a lower production cost than defatted BSF meal, as there is no expense involved in the fat extraction process. Wholemeal BSFL meal can be considered a leading ingredient to meet the emerging need for protein ingredients by the feed industry.

**Abstract:**

Black soldier fly (BSF, *Hermetia illucens*) larvae is a prominent ingredient in aquafeeds due to its high protein and energy contents. This study evaluated the digestibility of full-fat BSF larvae meal (FF-BSFL) and its inclusion in diets for tambaqui, *Colossoma macropomum*. The apparent digestibility coefficient of FF-BSFL for protein and energy was around of 88%, corresponding to 33.55% and 21.61 MJ kg^−1^ of digestible protein and energy, respectively. For the feeding trial, tambaqui juveniles (53.23 ± 1.07 g) were distributed in a completely randomized experimental design (n = 4; 150 L tanks; 10 fish per tank). Fish were fed diets including 0%, 5.25%, 10.50%, and 15.75% FF-BSFL to apparent satiation for 60 days. Fish fed 10.50% FF-BSFL dietary inclusion showed higher weight gain, feed intake, final biomass, and relative growth rate. The 10.50% FF-BSFL diet presented the highest index of economic profitability. Weight gain fitted a third-degree equation and the optimum FF-BSFL inclusion level was estimated at 11.6%. However, FF-BSFL dietary inclusion up to 15.75% did not impair growth fish performance. FF-BSFL seems to be a promising source of protein and energy for omnivorous fish aquafeed.

## 1. Introduction

The world population reached 8 billion people in 2022 and, according to the United Nations [1], the forecast is that it will reach 9 billion inhabitants in the next fifteen years. Increased life expectancy associated with advances in medicine, health and nutrition are among the reasons for this unprecedented population growth. Therefore, there is widespread concern about the emerging need to produce animal-based protein more efficiently and sustainably to meet the continuous growing human demand for food [2].

Insects are the most abundant macroscopic animals on the planet and represent 73% of the total described fauna [3]. The value of the insect market in 2023 is estimated at USD 2.6 billion. According to the Future Market Insights Global reports [4], the value of the insect market for animal nutrition worldwide is expected to increase at a compound annual growth rate of 20.9% from 2023 to 2033, reaching USD 17.4 billion by 2033. Insects as feed ingredients for aquafeeds are on the rise in science and industry sectors, since there is an emergent necessity for alternative protein sources to fish meal and fish oil [5].

Among the different species of insects, the black soldier fly (BSF, *Hermetia illucens* Linnaeus, 1758) has been proposed as one of the insects with the greatest potential as an ingredient for animal nutrition [6]. The production of BSF larvae (BSFL) is encouraged due to its characteristics that facilitate its handling, such as short development cycle, ability to live in high densities and take advantage of low-cost waste as substrate, fast growth, and low vulnerability to diseases, resulting in high survival rates [3,7].

BSFL can be processed in different ways which result in ingredients for the feed industry with different protein and fat contents. The defatted BSFL meal (D-BSFL) is produced by partial or total fat extraction using pressing or organic solvents, and the resulting defatted meal is then posteriorly dried and ground. D-BSFL has around 60% protein and 10–12% lipid content [8,9]. The full-fat BSFL (FF-BSFL) meal is easy to produce by drying and posteriorly grinding. The FF-BSFL is a low-cost technology when compared to D-BSFL as it avoids expenses associated with fat extraction processes. FF-BSFL has an average content of 42% crude protein and 30% lipids [10,11].

BSFL meal was evaluated positively when included or replacing fish meal in diets for Atlantic salmon (*Salmo salar* Linnaeus, 1758) [12], rainbow trout (*Oncorhynchus mykiss* Walbaum, 1792) [13], Nile tilapia (*Oreochromis niloticus* Walbaum, 1792) [14] and Siberian sturgeon (*Acipenser baerii* Brandt, 1869) [5]. The use of seven species of insects in the aquafeed industry was approved by the European Union (Regulation no. 2017/893), and it is a prominent issue in the world literature [15].

Tambaqui, *Colossoma macropomum* (Cuvier, 1818), is an omnivorous fish, native to the basins of the Amazon and Orinoco rivers and is one of the most reared species in South America and has already been introduced in the United States, China, Thailand, and the Philippines for aquaculture purposes [16,17]. In Brazil, tambaqui production reached 100.6 thousand tons in 2020 [18]. Tambaqui easily adapts to feed formulations and aquaculture production systems [16]; in addition, it can be reared with the replacement of 50% commercial feed for whole BSFL without impairing its growth performance [19]. Furthermore, tambaqui digests 86% of the crude protein of the defatted BSFL meal, and levels of up to 30% defatted BSFL meal can be used in its fishmeal-free diet [9]. However, there are no published data on the use of full-fat BSFL meal in tambaqui diets.

This study evaluated the inclusion of full-fat BSFL meal as an ingredient in tambaqui diets and the effects on nutrient and energy digestibility, growth performance, whole body composition, and economic analysis of diets.

## 2. Materials and Methods

The experiment was carried out at the Aquaculture Experimental Station of the Technology and Innovation Coordination**-**COTEI/INPA (3°05′26.7″ S and 59°59′41.1″ W), Amazon State, Brazil. The study complied with the standards required by the National Council for the Control of Animal Experimentation (CONCEA, 2018) and was approved by the Ethics Committee in Research on the Use of Animals of the National Institute for Research in the Amazon (INPA) with protocol no. 056/2017 and 010/2021.

### 2.1. Production and Processing of Full-Fat Black Soldier Fly Larvae Meal (FF-BSFL)

BSFL were produced in starter feed for broilers (crude protein: 21%; ether extract: 2.5%; crude fiber: 5%; moisture: 13%) as a substrate for standardizing its bromatological composition, and they were supplied by the Natuprotein company in the city of Manaus, Amazon, Brazil. BSFL were dehydrated in a forced air circulation oven at 55 °C for 24 h. Dehydrated larvae were ground in a multiprocessor with 0.8 mm mesh.

### 2.2. Digestibility Assay

A practical feed was formulated to meet the nutritional requirements of tambaqui and used as the reference diet (RD; Table 1). The test diet (TD) was obtained by replacing 30% of RD by FF-BSFL. The apparent digestibility coefficient (ADC) was determined by the indirect method using 0.5% chromium oxide III (Cr_2_O_3_), as a marker, in the experimental diets [20].

Tambaqui juveniles (48.66 ± 8.02 g) were housed in six conical tanks (200 L; n = 3; 17 fish per tank) fitted with a collection tube at the bottom for feces collection [21]. The tanks formed part of an open system with water renewal from an artesian well and with constant aeration. Fish were fed twice daily at a rate of 3% biomass for a 22-day period. Decanted feces were collected twice daily (8:00 am and 5:00 pm) and stored in a freezer (−20 °C) for later laboratory analyses.

The analyses to determine the chromium III oxide content in the samples of feeds and feces were carried out by the colorimetric method, according to the methodology described by Furukawa and Hiroko [22]. The standard curve was calculated from the nitro-perchloric digestion of the samples with known chromium III oxide concentrations. A reading was taken by a spectrophotometer, adjusted to a wavelength of 350 nm. The Apparent Digestibility Coefficients (ADCs) of nutrients and the energy of the RD and TD diets were calculated according to the following equation:

ADC (%) = 100 − (100 × ((%Chromium III in diet)/(%Chromium III in feces)) × ((% nutrient (or energy) in feces)/(% nutrient (or energy) in diet)).

The nutrient and energy ADCs of the FF-BSFL meal were calculated following the equation proposed by Bureau and Hua [23]:

ADCi (%) = ADCtd + (ADCtd-ADCref) × ((r × Nref)/(i × Ni))

where:

ADCi = the apparent digestibility coefficient of the ingredient;

ADCtd = the apparent digestibility coefficient of the TD nutrient;

ADCref = the apparent digestibility coefficient of the RD nutrient;

r = the proportion of the RD in the TD (0.65);

i = the proportion of the test ingredient in the TD (0.3);

Nref = the nutrient concentration in the RD (%);

Ni = the nutrient concentration in the test ingredient (%).

### 2.3. Feeding Trial

Four experimental diets were formulated according to the nutritional requirements of tambaqui juveniles [24,25], i.e., a control diet without inclusion of FF-BSFL meal (0%) and three diets with the following levels of inclusion of FF-BSFL meal: 5.25%, 10.50%, or 15.75% (Table 2). All ingredients were ground, sieved (0.8 mm), hydrated (25%), and extruded in a single-screw extruder (model MX-80, INBRAMAQ**^®^**; São Paulo, Brazil) with a 3 mm die.

Tambaqui juveniles (53.23 ± 1.07 g; 14.92 ± 0.47 cm) were distributed in a completely randomized experimental design, with four treatments and four replicates (150 L tanks; 10 fish/tank), totaling 16 experimental units. The experimental units formed part of a recirculation system with phytoremediation, constant aeration, and natural photoperiod. The fish were fed 3 times a day (10 am, 2 pm, 5 pm) until apparent satiation for 60 days.

### 2.4. Water Quality Parameters

The water quality parameters of dissolved oxygen, temperature, and pH were monitored daily using a multiparameter probe (PRO ODO, YSI**^®^**; Yellow Springs, OH, USA). The ammonia, nitrite, and alkalinity levels were monitored weekly using colorimetric and titrimetric kits (Alfakit AT 101; Alfakit, Florianópolis, Santa Catarina, Brazil).

### 2.5. Growth Performance and Biometric Indices

Fish biometrics were performed at the beginning, 30th day, and 60th day of the experiment to monitor the growth of the fish. Before the biometrics, fish were fasted for 24 h and anesthetized with a solution of 100 mg of benzocaine L^−1^ [26] to facilitate handling and avoid injuries to the fish.

At the end of the experiment, the growth performance data were expressed by the parameters obtained by the following calculations:average weight gain (g) = final weight–initial weight;biomass weight gain (kg) = final biomass–initial biomass;final biomass (g) = final number of fish × final average weight;feed consumption (g) = total weight of feed consumed per tank;apparent feed conversion = feed intake ÷ weight gain;relative growth rate (% day^−1^) = (e^g −1^) × 100;

where “e” is the Neperian number and g = (ln (final weight)-ln (initial weight))/(number of experimental days);
protein efficiency ratio (g ^g−1^) = biomass weight gain ÷ protein consumption;protein conversion efficiency (%) = [(final body weight × final body protein %) − (initial body weight × initial body protein %)/protein intake (g)] × 100;condition factor = 100 × (body weight/total length^3^).

Two fish from each experimental unit were anesthetized by immersion in 300 mg benzocaine L^−1^ until the loss of reflex activity and no reaction to external stimuli were noted. Afterward, fish were euthanized by spinal medulla rupture according to the rules of the CONCEA [27] and destined for the collection and weighing of viscera and liver to determine the following indices: viscerosomatic index = 100 × (viscera weight/body weight); hepatosomatic index = 100 × (liver weight/body weight).

### 2.6. Proximate Composition

The proximate compositions of the FF-BSFL meal, diets, feces, and whole-body fish were determined according to AOAC [28]. Moisture content was determined by oven drying at 105 °C until constant weight; total lipids by solvent (hexane) extraction for 6 h (Soxhlet); ash content by burning in a muffle furnace at 500 °C for 4 h; crude protein (CP) by the micro-Kjeldahl method and correcting the total nitrogen content by multiplying by the factor 6.25. Specifically, for the FF-BSFL meal sample, CP was determined by the micro-Kjeldahl method and by correcting the total nitrogen content by multiplying by the factor 5.6, as recommended by Janssen et al. [29]. Chitin was analyzed following the modified method of Abidin et al. [30] with washing, deproteinization (NaOH), demineralization (HCl), and drying of samples. The gross energy of samples was determined by an IKA 2600 calorimeter bomb (IKA**^®^**-Werke GmbH & Co. KG, Staufen, Germany).

### 2.7. Economic Analysis of Feeds

The cost of producing BSFL meal was determined through fixed capital investment for small-scale production of 24 kg of BSFL in a 21-day production cycle. Investment in civil construction was not considered. Production costs were determined based on the total operating cost, which is the sum of the effect operating cost (EOC) and the depreciation of fixed capital items. In the EOC, labor, equipment maintenance (20% of the new value per year), and the initial feed for chickens that served as substrate for the growth of the BSFL were added.

The labor cost was calculated based on the time used to produce the BSFL and the cost per hour worked (USD 1.67 h) in Brazil. The cost per hour worked was calculated based on a salary of USD 225.70 (equivalent to a minimum wage of BRL 1212.00 currently practiced in Brazil), plus 48% social security contributions for 200 h worked per month. Depreciation of equipment was calculated based on the straight-line method. The production of 1 kg of BSFL was estimated at USD 3.50.

The cost of the diet was calculated based on the market prices of the ingredients used (Table 2). The cost of diet processing was not considered. The analysis was based on local market retail prices converted to USD (USD 1 = BRL 5.37; exchange as of 3 December 2022). The diets cost USD 0.93/kg (0% FF-BSFL), USD 0.97/kg (5.25% FF-BSFL), USD 1.00/kg (10.50% FF-BSFL), and USD 1.02/kg (15.75% FF-BSFL).

The economic efficiency of the diets was assessed using input–output analysis [12,31]. To determine the relative economic efficiency and cost benefits of the tested diet per unit of fish gain, the economic conversion ratio (ECR) was calculated as fellows: feed cost/kg weight gain = (feed intake (g)/body weight gain (g)) × cost of feed (USD/1 kg). The economic profit index (EPI) was calculated using the formula from Stejskal et al. (2020): EPI (USD/fish) = (gain of body weight (kg) × live fish selling price (USD/1 kg) − (body weight gain (kg) × (feed cost (USD/1 kg)).

The calculation of profitability (PRO) used the balance between the selling price of fish and the cost of feed per kg of fish gain (ECR): PRO (USD/1 kg of fish gain) = sale price of live fish (USD/1 kg) − economic conversion ratio (USD/1 kg of fish gain). Economic efficiency was calculated using the following ratio: economic efficiency = profit per kg gain/feed cost per kg gain.

### 2.8. Statistical Analysis

The variables expressed as percentages were previously transformed by square root arcsine. Data were submitted to the Levene and Shapiro–Wilk tests to verify normality and homoscedasticity, respectively. If the data were not normally distributed, or had unequal variance, they were subjected to the nonparametric Kruskal–Wallis test, followed by the Dwass–Steel–Critchlow–Fligner test. The parametric data were submitted to one-way analysis of variance where diet (4 levels) was the factor. If there was a difference between treatments (diets), means were compared using the Tukey test at α = 0.05. The treatments were also treated as a continuous variable, and regression analysis was performed. If significant differences were found, a lack of fit test was performed to validate the regression. Variables were analyzed by regression using the CurveExpert Pro software, and models were selected for best fitting the data to the model by the coefficient of determination (r^2^) and by making a comparison by the F test and the AIC criterion.

## 3. Results

There were no significant differences in the water quality parameters during the experimentation period (Table 3). Tambaqui presented ADC above 88% for crude protein and energy of FF-BSFL, which resulted in 33.55% and 21.61 MJ kg^−1^ of digestible protein and energy, respectively. The ADC of lipids was 96.42% and resulted in 30.47% of digestible lipids for tambaqui. For chitin, tambaqui presented an ADC of 16.40%, which consisted of 1.72% digestible chitin (Table 4).

All groups of fish accepted the diets formulated with FF-BSFL and there was no record of mortality. Fish fed 10.5% FF-BSFL diet showed higher weight gain, final biomass, and relative growth rate compared to fish fed the other levels of dietary FF-BSFL (Table 4). The inflection point of the regression curve (Figure 1) was at 11.6% FF-BSFL dietary inclusion, which corresponded to 80.51 g of weight gain of tambaqui juveniles after 60 days of the experiment. Tambaqui fed 5.25% or 15.75% FF-BSFL dietary inclusion showed a similar growth performance to fish fed with the control diet (0% FF-BSFL). Apparent feed conversion, protein efficiency rate, protein conversion efficiency, and condition factor did not differ between experimental groups (Table 5).

There was no statistically significant difference in the proximate composition and protein conversion efficiency of tambaqui fed diets containing up to 15.75% FF-BSFL meal (Table 6).

The economic profit index values were higher (*p* < 0.05) for diets with 10.50% FF-BSFL meal inclusion. No statistical differences were observed for economic conversion ratio, profitability, and economic efficiency of feeds (Table 7).

## 4. Discussion

The water quality parameters of all experiments were within acceptable limits for tambaqui [32]. For new ingredients to be considered promising for aquafeed formulation, it is necessary to observe their nutrient and energy contents, in addition to their digestibility, attractiveness, effect on growth performance, and impact on animal health [33]. The nutritional value of FF-BSFL (42.3% crude protein; 37.9% corrected protein; 31.6% crude lipids) was within the average nutrient content found in the literature. In a review article, Tran et al. [34] found that the composition of BSFL ranged from 41% to 44% for crude protein and from 15% to 34% for total lipids. The nutritional composition of BSFL can be influenced by the nutritional variation of the substrate on which the insect larvae grew [7]. In addition, the BSFL crude protein value can be overestimated if the factor 5.6 is not used to correct chitin nitrogen or non-protein nitrogen [29].

The values of ADCs of the protein and energy of the FF-BSFL meal (higher than 88%) were similar to the values reported for Nile tilapia (*Oreochromis niloticus*) juveniles (70.0–82.1%) fed different insect meals [35]. Chitin can inhibit the absorption of lipids in the intestine and increase their excretion [36]. However, this was not observed in this study, since the FF-BSFL lipid ADC was greater than 95%. The low ADC values of chitin suggest that tambaqui has a low chitinase activity, as demonstrated in Nile tilapia, another neotropical freshwater species [34]. However, it is possible that a prolonged period of feeding FF-BSFL-based diets collaborates to increase the digestibility of chitin, since the digestive enzymes of tambaqui can specialize in digesting this carbohydrate [37]. In nature, tambaqui consume a diversified diet and, like most omnivorous fish, they have morphological adaptations such as numerous pyloric caeca and plasticity of digestive enzymes that contribute to the better utilization of nutrients [38,39].

Extrusion processing of feeds can make complex carbohydrates, such as chitin, more available. In fact, extrusion cooking can break down complex polysaccharides into smaller components, such as mono- or disaccharides, which are more digestible. The levels of chitin present in the experimental diets did not affect the growth performance of tambaqui, as was also observed in Atlantic salmon fed with FF-BSFL meal and paste of up to 12.5% dietary inclusion [40].

The higher consumption of the 10.50% FF-BSFL diet reflected better growth performance results for tambaqui. Insects have pheromones on their surface, known as “natural attractants”, which may have acted as a palatability agent in the experimental diets [41,42]. However, the inclusion level of 15.75% may indicate a limit to the palatability of FF-BSFL dietary inclusion or fish satiety due to the chitin content.

Chitin is a carbohydrate resistant to chemical degradation, but it presents an energy content of approximately 17.1 kJ g**^−1^** [43]. This fraction of energy was accumulated in diets with increasing levels of FF-BSFL inclusion; however, as chitin digestibility was low (16.40%), it may have been a kind of filler with low digestible energy content [44]. This may have kept the fish satiated for a longer period of time, influencing their feed intake and weight gain. Similarly, Siberian sturgeon (*Acipenser baerii*) fed FF-BSFL meal-based diets showed improvements in growth performance parameters [5].

In aquaculture, feeding costs amount to more than half of production costs [45]. The reduction in fish feeding costs contributes to greater profitability and economical sustainability of fish farming. Studies on the effects of using ingredients from BSFL production in diets for neotropical fish related to the economic analysis are still limited. In this study, the improvement in feed conversion of fish that consumed 10.50% FF-BSFL dietary inclusion indicates a contribution of the use of FF-BSFL meal to improve the efficiency of tambaqui production.

Despite the feed price increasing according to the inclusion of FF-BSFL meal, feeding tambaqui with 10.50% FF-BSFL meal diet increased the economic profit index by an average of 6% in relation to the diet with 0% FF-BSFL meal. In the case of Nile tilapia fed diets with 75% replacement of fish meal by BSFL meal, an increase of 3.97% in the economic profit index was observed [46], and greater economic efficiency with replacement of 100% fish meal was found [31]. When diets with FF-BSFL were tested in sturgeon, the economic profit index increased according to the level of inclusion [5]. In these studies, the price of BSFL meal was the main factor that had an impact on the variation in diet prices.

There is a worldwide tendency to produce BSFL on a large scale, which can reduce the costs of production, positively impacting the economic efficiency rates of feeds formulated with BSF ingredients. Furthermore, it must be considered that full-fat BSFL meal has a lower production cost than the defatted BSF meal, as there is no expense associated with the fat extraction process. The optimization of growth performance of tambaqui fed FF-BSFL meal-based diets and the economic profit index of its production indicate the possibility of more sustainable aquaculture, which can strengthen the bioeconomy around the world to produce more fish as food for humans.

## 5. Conclusions

Full-fat black soldier fly larvae meal is well digested by tambaqui and the inclusion of 11.6% FF-BSFL in diet yielded the maximum weight gain in tambaqui juveniles. Based on the results obtained, FF-BSFL meal can be considered as a prominent ingredient to meet the emerging need for protein sources by the aquafeed industry.

## Figures and Tables

**Figure 1 animals-13-00360-f001:**
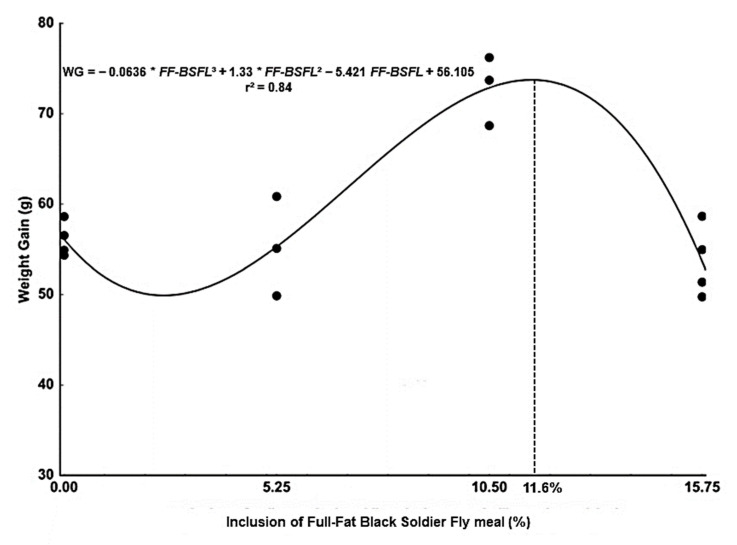
Weight gain of tambaqui juveniles fed diets containing increasing levels of full-fat black soldier fly (FF-BSFL) meal.

**Table 1 animals-13-00360-t001:** Formulation and chemical composition of the reference diet utilized in the digestibility assay (values expressed on dry matter basis).

Ingredients	Reference Diet (g 100 g^−1^)
Soybean meal	53.22
Corn meal	31.34
Wheat middlings	6.27
Meat and bone meal	4.98
Soybean oil	1.99
Vit and Min premix ^a^	1.00
Dicalcium phosphate	0.50
Common salt	0.10
DL-methionine	0.08
BHT	0.02
Chromium oxide III	0.50
**Chemical composition and Energy**	
Dry matter (%)	93.19
Crude protein (%)	31.00
Crude lipids (%)	5.77
Crude fiber (%)	1.85
Ashes (%)	8.49
Gross energy (MJ kg^−1^)	14.11

^a^ Vitamin and mineral mix (Nutron^©^; Campinas, SP, Brazil) per kg of product: folic acid (250 mg), pantothenic acid (5000 mg), antioxidant (600 mg), biotin (125 mg), cobalt (25 mg), copper (2000 mg), iron (13,820 mg), iodine (100 mg), manganese (3750 mg), niacin (5000 mg), selenium (75 mg), vitamin A (1,000.000 IU), vitamin B1 (1250 mg), vitamin B12 (3750 mg), vitamin B2 (2500 mg), vitamin B6 (2485 mg), vitamin C (28,000 mg), vitamin D3 (500,000 IU), vitamin E (28,000 IU), vitamin K3 (500 mg), zinc (17,500 mg).

**Table 2 animals-13-00360-t002:** Formulation and composition of the experimental diets (on dry matter basis).

Ingredients (%)	Diets			
	0%	5.25%	10.5%	15.75%
Full-fat BSFL meal	-	5.25	10.50	15.75
Soybean meal	53.50	51.30	48.00	46.32
Corn meal	31.50	31.00	29.75	30.30
Wheat middlings	6.30	4.00	3.90	1.50
Meat and bone meal	5.00	5.00	5.00	5.00
Soy oil	2.00	2.00	1.70	-
Vit and Min premix ^a^	1.00	1.00	1.00	1.00
Dicalcium phosphate	0.50	0.28	-	-
Common salt	0.10	0.10	0.10	0.10
DL-methionine	0.08	0.05	0.03	0.01
BHT	0.02	0.02	0.02	0.02
**Chemical composition and Energy**				
Dry matter (%)	93.19	92.84	91.26	91.87
Crude protein (%)	31.0	30.02	30.4	33.5
Corrected protein ^b^ (%)	31.0	27.30	27.86	28.91
Crude lipids (%)	5.77	4.16	4.44	4.86
Crude fiber (%)	1.85	1.44	1.39	1.58
Ashes (%)	8.49	7.72	5.71	6.81
Chitin (%)	-	0.55	1.10	1.65
Gross energy (MJ kg^−1^)	14.12	14.04	13.97	14.63

^a^ Vitamin and mineral mix (Nutron^©^; Campinas, SP, Brazil) per kg of product: folic acid (250 mg), pantothenic acid (5000 mg), antioxidant (600 mg), biotin (125 mg), cobalt (25 mg), copper (2000 mg), iron (13,820 mg), iodine (100 mg), manganese (3750 mg), niacin (5000 mg), selenium (75 mg), vitamin A (1,000.000 IU), vitamin B1 (1250 mg), vitamin B12 (3750 mg), vitamin B2 (2500 mg), vitamin B6 (2485 mg), vitamin C (28,000 mg), vitamin D3 (500,000 IU), vitamin E (28,000 IU), vitamin K3 (500 mg), zinc (17,500 mg). ^b^ Corrected crude protein was calculated by applying a nitrogen-to-protein conversion factor of Kp = 5.6.

**Table 3 animals-13-00360-t003:** Water quality parameters in rearing tambaqui fed diets containing full-fat black soldier fly larvae (FF-BSFL) meal during the experimental period.

Diets	Dissolved Oxygen (mg L^−1^)	Temperature (°C)	pH	Ammonia (mg L^−1^)	Nitrite (mg L^−1^)	Alkalinity (mg L^−1^)
Digestibility assay						
Reference diet	6.60 ± 0.01	26.47 ± 0.06	5.83 ± 0.06	0.10 ± 0.05	-	-
Test diet	6.45 ± 0.03	26.43 ± 0.06	5.87 ± 0.06	0.11 ± 0.04	-	-
*p*-value	0.784	0.519	0.520	0.471	-	-
Feeding trial						
0%	6.10 ± 0.04	27.85 ± 0.05	6.74 ± 0.05	0.91 ± 0.11	0.65 ± 0.02	49.7 ± 3.46
5.25%	6.08 ± 0.04	27.90 ± 0.03	6.73 ± 0.04	0.82 ± 0.28	0.64 ± 0.03	50.5 ± 3.10
10.50%	6.12 ± 0.04	27.91 ± 0.02	6.68 ± 0.01	0.78 ± 0.26	0.67 ± 0.03	50.5 ± 3.10
15.75%	6.10 ± 0.04	27.90 ± 0.03	6.69 ± 0.04	0.88 ± 0.31	0.65 ± 0.02	49.9 ± 3.5
*p*-value	0.999	1.00	0.999	0.390	0.891	0.725
Regression	NS	NS	NS	NS	NS	NS

**Table 4 animals-13-00360-t004:** Composition, apparent digestibility coefficient (ADC), values of digestible nutrients, and energy of the full-fat black soldier larva meal (FF-BSFL).

Nutrient/Energy	Crude Composition of FF-BSFL Meal	ADC (%)	Digestible Nutrients and Energy
Dry matter (%)	96.80 ± 0.28	80.82 ± 6.25	78.23 ± 0.23
Crude protein (%)	42.30 ± 0.42	88.53 ± 3.09	37.45 ± 0.38
Corrected protein (%) ^a^	37.90 ± 1.40	88.53 ± 2.26	33.55 ± 1.20
Crude lipids (%)	31.60 ± 0.42	96.42 ± 0.41	30.47 ± 0.41
Ashes (%)	10.50 ± 0.00	66.69 ± 3.10	7.00 ± 0.00
Chitin (%)	10.50 ± 0.46	16.40 ± 6.56	1.72 ± 0.08
Gross energy (MJ kg^−1^)	24.29 ± 0.08	88.98 ± 3.37	21.61 ± 0.07

^a^ Corrected crude protein was calculated by applying a nitrogen-to-protein conversion factor of Kp = 5.6.

**Table 5 animals-13-00360-t005:** Growth performance and biometric indices of tambaqui fed diets containing graded levels of full-fat black soldier fly larvae (FF-BSFL) meal for 60 days.

Diets	WG (g)	FB (g)	FC (g)	AFC	RGR (%)	PER (%)	PCE (%)	CF	VI (%)	HI (%)
**0%**	56.10 ± 1.92 ^b^	533.57 ± 65.26 ^b^	692.09 ± 49.80 ^b^	1.31 ± 0.11	1.24 ± 0.02 ^b^	2.95 ± 0.48	63.21 ± 3.74	1.77 ± 0.09	7.08 ± 1.03	2.08 ± 0.29
**5.25%**	55.27 ± 5.48 ^b^	552.71 ± 54.80 ^b^	791.95 ± 2.11 ^ab^	1.44 ± 0.15	1.21 ± 0.09 ^b^	2.58 ± 0.66	44.39 ± 10.23	1.77 ± 0.01	7.61 ± 1.01	2.26 ± 0.50
**10.50%**	72.88 ± 3.83 ^a^	728.80 ± 38.30 ^a^	917.25 ± 40.09 ^a^	1.26 ± 0.01	1.47 ± 0.05 ^a^	2.89 ± 0.40	64.96 ± 12.03	1.80 ± 0.07	7.25 ± 0.78	1.92 ± 0.19
**15.75%**	53.69 ± 3.96 ^b^	510.76 ± 57.63 ^b^	662.51 ± 88.92 ^b^	1.30 ± 0.04	1.18 ± 0.09 ^b^	2.97 ± 0.40	39.44 ± 6.36	1.73 ± 0.06	7.93 ± 0.55	2.13 ± 0.27
*p*-value	<0.001 *	0.002 *	<0.001 **	0.232	<0.001 *	0.719 *	0.378	0.595 *	0.241	0.350
*Regression*										
*p*-value	<0.001	0.003	0.022	0.046	<0.001					
r^2^	0.84	0.75	0.79	0.40	0.79					
Model	Cubic	Cubic	Cubic	Cubic	Cubic	NS	NS	NS	NS	NS

WG: weight gain; FB: final biomass; FC: feed consumption; AFC: apparent feed conversion; RGR: relative growth rate; PER: protein efficiency rate; PCE: protein conversion efficiency; CF: condition factor. Different letters in column indicate significant difference. (*) Data were analyzed using one-way ANOVA, when its assumptions were validated by Shapiro–Wilk for normal distribution and Levene’s test for homogeneity of variance. If significant differences were detected (*p* < 0.05), then data were subjected to a Tukey HSD test. If the data were not normally distributed, or had unequal variance, they were subjected to the nonparametric Kruskal–Wallis test, followed by the Dwass–Steel–Critchlow–Fligner test (**). The treatments were also treated as a continuous variable, and regression analysis was performed. If significant differences were found, a lack of fit test was performed to validate the regression. The best-fitting model was chosen by AIC criteria and F test based on the highest r^2^ and lowest *p*-value.

**Table 6 animals-13-00360-t006:** Proximate composition of whole body of tambaqui fed diets containing graded levels of full-fat black soldier fly larvae (FF-BSFL) meal for 60 days.

Diets	Moisture	Crude Protein	Crude Lipids	Ashes
**0%**	8.02 ± 0.47	50.07 ± 3.72	26.00 ± 0.62	15.52 ± 1.17
**5.25%**	7.24 ± 1.02	45.57 ± 0.38	25.79 ± 0.42	14.71 ± 0.33
**10.50%**	7.85 ± 0.71	48.82 ± 3.28	26.02 ± 0.73	15.61 ± 0.81
**15.75%**	8.48 ± 0.77	47.48 ± 3.75	25.90 ± 0.53	15.36 ± 0.58
*p*-value	0.244	0.361	0.957	0.428
*Regression*	NS	NS	NS	NS

Data were analyzed using one-way ANOVA, after its assumptions were validated by Shapiro–Wilk for normal distribution and Levene’s test for homogeneity of variance. The treatments were also treated as a continuous variable, and regression analysis was performed. If significant differences were found, a lack of fit test was performed to validate the regression.

**Table 7 animals-13-00360-t007:** Economic analysis of feeds with the different inclusion levels of full-fat BSFL meal in the diets.

Diets	ECR	EPI	PRO	Economic Efficiency
**0%**	1.21 ± 0.11	0.30 ± 0.04 ^ab^	0.28 ± 0.11	0.23 ± 0.11
**5.25%**	1.40 ± 0.14	0.29 ± 0.03 ^ab^	0.09 ± 0.14	0.07 ± 0.11
**10.50%**	1.26 ± 0.01	0.36 ± 0.02 ^a^	0.23 ± 0.01	0.18 ± 0.01
**15.75%**	1.32 ± 0.04	0.24 ± 0.03 ^b^	0.17 ± 0.04	0.13 ± 0.04
*p*-value *	0.099	0.003	0.099	0.124
*Regression*				
*p*-value	0.042	0.005	0.043	
r^2^	0.45	0.73	0.45	
Model	Cubic	Cubic	Cubic	NS

ECR: economic conversion ratio (USD/1 kg of fish gain); EPI: economic profit index (USD/fish); PRO: profitability (USD/1 kg of fish gain). Different letters in column, indicate significant difference. (*) Data were analyzed using one-way ANOVA, when its assumptions were validated by Shapiro–Wilk for normal distribution and Levene’s test for homogeneity of variance. If significant differences were detected (*p* < 0.05), then data were subjected to a Tukey HSD test. The treatments were also treated as a continuous variable, and regression analysis was performed. If significant differences were found, a lack of fit test was performed to validate the regression. The best-fitting model was chosen by AIC criteria and F test based on the highest r^2^ and lowest *p*-value.

## Data Availability

Data from the study are available from the corresponding authors upon reasonable request.

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
