# Peer review of "Full-Fat Black Soldier Fly Larvae Meal in Diet for Tambaqui, Colossoma macropomum: Digestibility, Growth Performance and Economic Analysis of Feeds"

_animals, 2023, doi:10.3390/ani13030360_

Round 1

Reviewer 1 Report

The study revealed really important and interesting data for aquaculture. I have just some minor recommendations: 

L50: At the first mention of a species, authors and dates must also be mentioned.

L63-68: I recommend the same as above. 

In material & methods, the authors added the water quality parameters. I recommend removing the values from here and making a subchapter in the results to present these values. Also, statistical tests should be carried out to assess significant differences among the water quality parameters of tanks.

L176-185: every equation should be presented in a different row. In this case, the equations are not transparent. 

Fig1.: The diagram should be revised, in this case, the equation of the curve is hardly visible. 

Author Response

REVIEWER #1          

The study revealed really important and interesting data for aquaculture. I have just some minor recommendations: 

L50: At the first mention of a species, authors and dates must also be mentioned.

Authors (AUTH). Thank you for your suggestion. The text was modified accordingly.

L63-68: I recommend the same as above. 

AUTH. Thank you for your suggestion. The text was modified accordingly.

In material & methods, the authors added the water quality parameters. I recommend removing the values from here and making a subchapter in the results to present these values. Also, statistical tests should be carried out to assess significant differences among the water quality parameters of tanks.

AUTH. Thank you for your correction. We added a subchapter in the Results section to present water quality parameters, as requested.

L176-185: every equation should be presented in a different row. In this case, the equations are not transparent. 

AUTH. Thank you for your suggestion. The text was modified accordingly.

Fig1.: The diagram should be revised, in this case, the equation of the curve is hardly visible.

AUTH. Thank you for your suggestion. We hope to have addressed your concerns and to improve the visibility of the curve.

Reviewer 2 Report

1, Grammar errors should be corrected, such as line 335, 377, etc.

2, The current status of utilization of this insect protein in feed industry and more details of the characteristics of this feed ingredient should be included in the introduction section.

3, To evaluate the performance of a new protein ingredient, only growth and digestibility is not enough. We need to make sure fish can grow with healthy state in quite long period. Therefore, I am not sure whether the authors have detected the parameters related to health, if so, the inclusion of parameters reflecting fish health will be useful for readers to evaluate the possibility of this new protein application in a more complete view.

Author Response

REVIEWER #2                                 

1, Grammar errors should be corrected, such as line 335, 377, etc.

AUTH. Thank you for your correction. The text was modified accordingly.

2, The current status of utilization of this insect protein in feed industry and more details of the characteristics of this feed ingredient should be included in the introduction section.

AUTH. Thank you for your suggestion. We added the information requested in Introduction section.

3, To evaluate the performance of a new protein ingredient, only growth and digestibility is not enough. We need to make sure fish can grow with healthy state in quite long period. Therefore, I am not sure whether the authors have detected the parameters related to health, if so, the inclusion of parameters reflecting fish health will be useful for readers to evaluate the possibility of this new protein application in a more complete view.

AUTH. We apologize to Reviewer 2 as we do not have blood parameter data for this experiment.

Reviewer 3 Report

Review animals-2142591

The work was oriented to examine the effects of full-fat black soldier fly larvae (FF-BSFL) meal inclusion in diets on the digestibility, growth performance, whole-body composition of tambaqui (Colossoma macropomum) and its economic viability. Data indicated higher weight gain, final biomass and relative growth rate in fish fed 10.5% FF-BSFL when compared to fish fed the other levels of dietary FF-BSFL. This higher growth was accompanied by a higher economic profit index value.

This research complements other studies on tambaqui by revealing new knowledge about the inclusion of black soldier fly larvae inclusion in diets. This study represents an important contribution to the sustainability of the aquaculture sector. This manuscript is well written and straight forwarded. However, there are some minor remarks that the authors should address during the revision process.

Minor revisions

Line 31: I would suggest replacing “10 fish tank-1” per “10 fish per tank” or “10 fish/tank”. In this context is clearer and more correct.

Line 31: Why authors have chosen the mentioned range of concentrations of FF-BSFL inclusion (5.25%, 10.50% and 15.75%)?

Section 2.4 (line 162): Authors indicate the average of ammonia levels acquired from all tanks during the experimental period. Did authors see significant differences in ammonia values between treatments in the feeding trial?

- Section 2.6 (line 193): Authors should indicate the number of fish used for proximate composition determination.

- Line 342: reference 41 is mentioned twice. It should be corrected.

- The reference 40 is not mentioned in the text. It should be corrected.

Author Response

REVIEWER #3

The work was oriented to examine the effects of full-fat black soldier fly larvae (FF-BSFL) meal inclusion in diets on the digestibility, growth performance, whole-body composition of tambaqui (Colossoma macropomum) and its economic viability. Data indicated higher weight gain, final biomass and relative growth rate in fish fed 10.5% FF-BSFL when compared to fish fed the other levels of dietary FF-BSFL. This higher growth was accompanied by a higher economic profit index value.

This research complements other studies on tambaqui by revealing new knowledge about the inclusion of black soldier fly larvae inclusion in diets. This study represents an important contribution to the sustainability of the aquaculture sector. This manuscript is well written and straight forwarded. However, there are some minor remarks that the authors should address during the revision process.

Minor revisions

-Line 31: I would suggest replacing “10 fish tank-1” per “10 fish per tank” or “10 fish/tank”. In this context is clearer and more correct.

AUTH. Thank you for your correction. The text was modified accordingly.

-Line 31: Why authors have chosen the mentioned range of concentrations of FF-BSFL inclusion (5.25%, 10.50% and 15.75%)?

AUTH. The dietary FF-BSFL inclusions were chosen so as not to extrapolate the fat content of the diets for tambaqui.

-Section 2.4 (line 162): Authors indicate the average of ammonia levels acquired from all tanks during the experimental period. Did authors see significant differences in ammonia values between treatments in the feeding trial?

AUTH. No differences in ammonia levels were observed between the experimental tanks due to the biologic filter of the water recirculation system.

- Section 2.6 (line 193): Authors should indicate the number of fish used for proximate composition determination.

AUTH. The information as added in the text.

- Line 342: reference 41 is mentioned twice. It should be corrected.

AUTH. Thank you for your correction. The text was modified accordingly.

-The reference 40 is not mentioned in the text. It should be corrected.

AUTH. Thank you for your correction. The reference has been added in the revised version of the text.

Round 2

Reviewer 2 Report

The authors answered the most of my questions, but still one is left due to the experimental design. I suggest the authors make a careful consideration for the next research project.